# Estimation of factors contributing to level differences in Japanese university decathlon athletes

Yuki Ashino[1,2]*, Yasushi Ikuta[3], Noriyuki Kida[2]

**1** Department of Biotechnology, Graduate School of Science and Technology, Kyoto Institute of Technology, Kyoto, Japan, **2** Department of International Engagement & Information Department, Japan Sport Council, Tokyo, Japan, **3** Department of Sports Sciences, Osaka Kyoiku University, Osaka, Japan

* d3841001@edu.kit.ac.jp

**Data Availability Statement:** All data files are available from figshare (https://figshare.com/projects/Estimation_of_Factors_Contributing_to_Level_Differences_in_Japanese_University_Decathlon/174495).

## Abstract

This study aimed to estimate the factors that cause differences in competition level based on the competition performance structure among university student decathletes in Japan. The results of factor analysis using the maximum likelihood method (Oblimin rotation), assuming a five-factor structure estimated from parallel analysis and the information criterion, revealed the following competitive performance structures: "running speed and body projection," "running endurance," "rotational throwing power," "translational throwing power," and "vertical leaping power." Some of these were similar to the competitive performance structures of the world's top athletes, but they were found to have a unique structure: two throwing powers. The analysis results using latent rank theory allowed us to evaluate them on a seven-point scale. The calculated item reference profile, difficulty index, and discrimination index for each rank indicated that shot put and 100 and 400 m performance formed the basis of decathlon performance. In addition, high jump and pole vault, which fall under the category of "vertical leaping power," retained high difficulty and discrimination and were revealed to affect the stage of achievement of overall performance in the decathlon.

## Introduction

The decathlon is an athletic competition in which male athletes compete in a combined event over two days. Athletes compete in 10 events, and the results are converted into points using a conversion formula. Therefore, besides physical ability, athletes must have the athletic skills to perform well in all events.

The structure of athletic skills required to demonstrate performance in the decathlon (hereinafter referred to as "competition performance structure") is often classified into three categories: "running," "jumping," and "throwing". However, they often conflate distinct physiological and kinematic factors. For example, "running" includes short-distance running events such as 100 m, which requires strength of the lower limb muscle groups and rapid force generation, and 1500 m, which requires predominantly endurance. Most research reports on the performance structure of decathlon events indicate that the structure of short-distance running events, such as 100 and 1500 m, differs [1–8]. Therefore, it can be inferred that the three

**Funding:** The author(s) received no specific funding for this work.

**Competing interests:** The authors have declared that no competing interests exist.

structures of "running," "jumping," and "throwing" are not sufficient for evaluating an athletes performance.

The performance structure of competitions has been analyzed in terms of inferential statistics using factor analysis. Linden [9] conducted a factor analysis on 139 decathletes who competed in eight Olympic Games since WWII, utilizing their discipline-specific records. The analysis yielded a diagonal solution. As a result, he reported that the decathlon consists mainly of four factors: 1. Running speed, consisting of short-distance running events; 2. Explosive arm strength or object projection, consisting of throwing events; 3. Running endurance, consisting of long-distance running events; 4. Explosive leg strength or body projection, consisting of jumping events. Moreover, Wimmer et al. [6] reported a four-factor structure similar to that reported by Linden [9]. They achieved this by conducting a similar factor analysis on the world's top decathletes from 1998 to 2009, while Kondo [10] performed a similar analysis on decathletes from 1985 to 2006. Conversely, Fan [3] conducted an analysis that encompassed 14 top Chinese athletes and 14 elite athletes from around the world. In their study, four factors were extracted. Notably, the 1500m (long-distance running) and high jump (jumping) were combined into a single factor, while pole vault was identified as a separate, distinct factor. Park et al. [4], who studied Olympic athletes from 1988 to 2008, reported one factor for the short-distance running event, one for the throwing and jumping events, and one for 1500m. As described above, some competition performance structures described in previous studies are similar, while others differ significantly. This reflects the different relationships among variables within the target population. The estimation methods used in previous studies primarily involve principal component analysis and factor analysis, which are analytical methods that compress dimensions or extract standard components based on the correlation coefficients of many variables. In addition, when considering the previously discussed factor analysis and estimation methods used in various studies, it's worth noting that there can be substantial variations in the transverse area of the psoas muscles across [11]. This morphological aspect adds another layer of complexity to the analysis of factors and components related to athletic performance in decathletes. Furthermore, Van Damme et al. [12] warn that estimating the population hides the characteristics of samples with values outside the mean. More precisely, within the context of the decathlon, the competitive landscape can, to a certain extent, mirror the overall competitiveness within national athletics [5]. Therefore, it is considered necessary to strictly specify the population of athletes to be evaluated, the competition level, race, country, and age group to which the athletes in the sample belong, before estimating the competition performance structure that is being assessed.

Van Damme et al. [12] speculate that, considering "the principle of allocation" in order to achieve excellence in a particular discipline is detrimental to overall decathlon performance, i.e., decathletes are expected to perform well in all disciplines and to be a generalist. Cox & Dunn [2] also stated that there is no benefit in being a specialist in a particular discipline. Nonetheless, they tentatively conclude that "the decathlon favors athletes who excel in field events." This suggests that there is a superiority or inferiority between disciplines according to the level of competition, even when aiming to become a "generalist." Estimating this superiority or inferiority may improve decathlon performance more efficiently.

In recent years, Japanese track and field athletes have been producing athletes who can compete with the world's best in a variety of events, and the sport has shown remarkable growth. However, it is still difficult for athletes to break the participation standard record in the decathlon. In Japan, the decathlon is held for university students, and the results are readily available online, but there are few reports on the performance of the athletes. Therefore, this study aimed to estimate the factors influencing the differences in performance structure and performance levels among Japanese university student decathletes.

## Materials and methods

### Subjects

The analysis participants were athletes who competed in the decathlon at the regional student championships in athletics and the Japan National University Championships in Athletics held from 2003 to 2022. Data for this study were obtained from the Inter-University Athletics Union of Japan (https://www.iuau.jp/index.html). The variables collected in this study were athlete name, year, region, total points, and event-specific records and conversion scores. The event-specific competition records were used to estimate the competition performance structure, and the event-specific conversion scores and total points were used to estimate the competition level. In the collection of competition results, some of the athletes had competed for multiple years, so there was some duplication of the same athletes. Therefore, we referred to the total points of individual athletes, and the dataset with the highest value was included in the analysis. Furthermore, after removing duplicates, all names were removed in order to blind the analysis. As a result, 807 athletes were included. Descriptive statistics for the participants involved in this study are shown in Fig 1 and Table 1. The lower triangular matrix in Fig 1 shows the scatter plots; the upper panel shows the correlation coefficients; the diagonal line shows the histograms for each field.

### Data analysis

**Exploratory factor analysis.** The purpose of this section was to estimate the competition performance structure. First, in order to find a reasonable factor structure, we used the records of the 10 events as variables and conducted the parallel analysis using the maximum likelihood method. Akaike information criterion (AIC), Bayesian information criterion (BIC), and root mean square error of approximation (RMSEA) were used to estimate the number of factors. Then, a factor analysis using the maximum likelihood method was conducted based on the estimated number of factors. Studies that have applied factor analysis [9, 10] have employed the orthogonal solution to estimate the competition performance structure. Therefore, the number of factors considered by the factor analysis of the maximum likelihood method was estimated to obtain the oblique solution. The RMSEA and Tucker-Lewis index (TLI) were used as goodness of fit indices.

**Latent rank theory.** This section aimed to estimate the factors contributing to the difference in competition level by using 11 variables, including 10 competition scores and total points, to create a classification of competition levels. Latent Rank theory (LRT) [13] is a non-parametric test theory that uses the self-organizing map (SOM) or generative topographic mapping (GTM) mechanism to estimate item response analysis [14].

LRT is a method that allows participants to be divided into multiple groups, similar to cluster analysis, latent class analysis, and latent profile analysis. LRT differs from these clustering methods in that the classified groups are ordered. This theory allows us to screen the levels of competition in the decathlon with a certain degree of confidence and to infer the factors that act to discriminate the difficulty and level of competition in each event.

### Analysis software

In this study, factor analysis was performed using R version 4.1.2 [15] and R Studio version 9.1.372 [16]. The main packages used were psych [17] for parallel analysis, calculation of information criterion, and factor analysis and effsize [18] for effect size calculation. In addition, HAD [19], a GUI-based free software, was used for LRT with continuous variables.

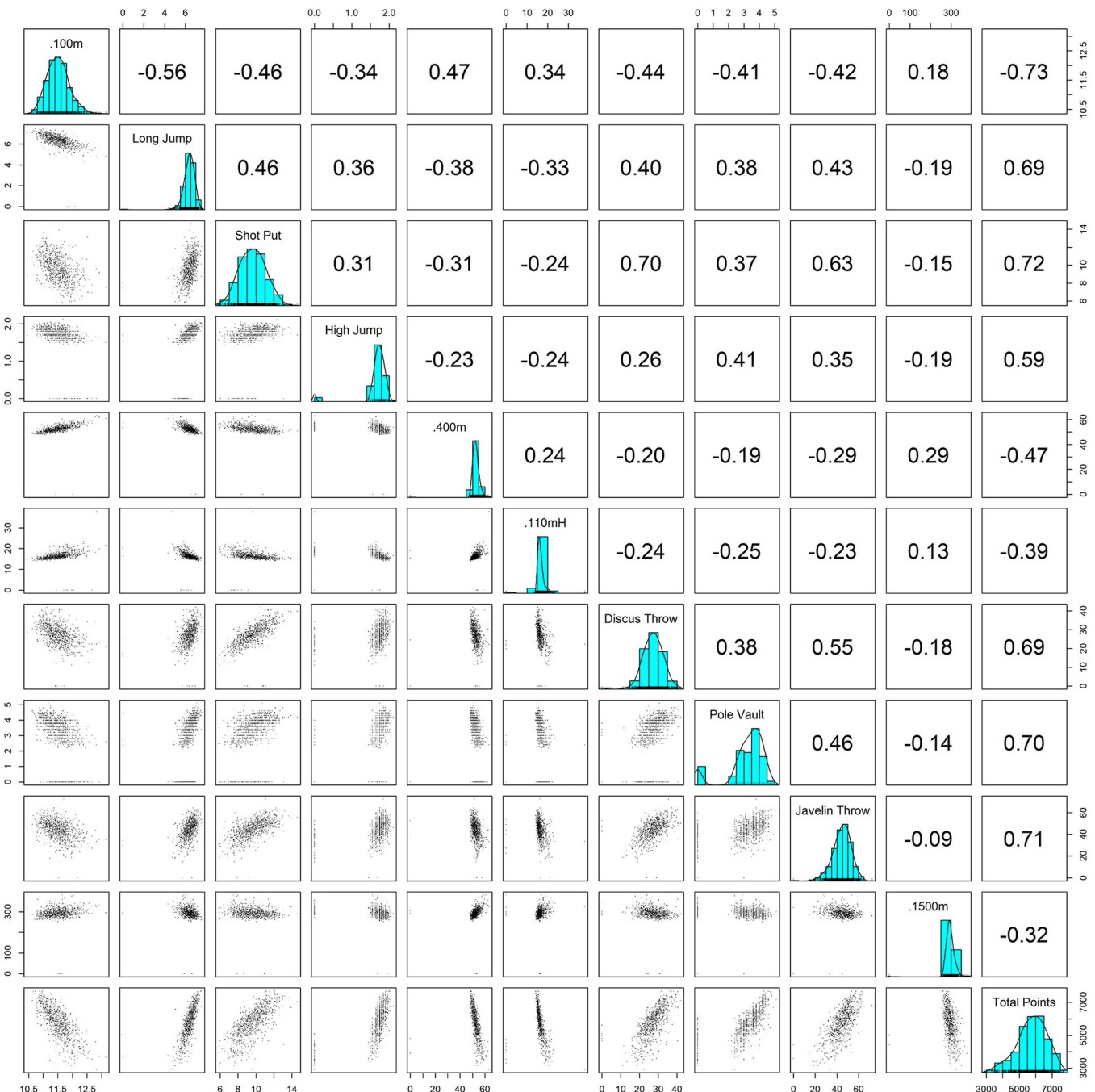

**Fig 1. Distribution and correlation of decathlon raw data.**

## Results

### Exploratory factor analysis

In performing the factor analysis, the Kaiser–Meyer–Olkin sampling adequacy criterion was calculated and compared to the evaluation criteria proposed by Kaiser [20], which confirmed that the data set was good (Kaiser-Meyer-Olkin criterion = 0.85, Measures of sampling adequacy = 0.76–0.93).

**Table 1. Sample characteristics.**

| Variables | Unit | N | | Mean(SD) | |
|---|---|---|---|---|---|
| | | Recorded | No Record[a] (point = 0) | Raw | Point |
| 100 m | s | 807 | 0 | 11.51(0.38) | 752.1(78.3) |
| Long Jump | m | 804 | 3 | 6.34(0.61) | 666.7(112.0) |
| Shot Put | m | 807 | 0 | 9.62(1.42) | 464.0(85.3) |
| High Jump | m | 775 | 32 | 1.68(0.36) | 560.6(150.1) |
| 400 m | s | 805 | 2 | 52.56(3.55) | 696.4(105.6) |
| 110 m Hurdle | s | 798 | 8 | 16.41(2.30) | 673.4(159.5) |
| Discus Throw | m | 801 | 6 | 27.21(5.58) | 412.3(104.4) |
| Pole Vault | m | 726 | 84 | 3.17(1.20) | 446.4(209.2) |
| Javelin Throw | m | 805 | 2 | 44.42(8.53) | 509.3(122.3) |
| 1500 m | s | 804 | 3 | 294.54(25.85) | 587.2(110.8) |
| Total Points | point | 807 | 0 | - | 5766.5(903.2) |

[a]No record include DNS, DNF, NM, and DQ.

A parallel analysis using the maximum likelihood method resulted in the scree plot shown in Fig 2. Up to the fourth factor, the information content of the original data was more significant than that of the random data. Therefore, the parallel analysis indicated that a three-factor structure was appropriate for estimation. Next, the BIC and AIC, which are information criterion, and RMSEA, a goodness-of-fit index, were checked, as shown in Table 2. The BIC showed a minimum value when the number of factors was three, while the AIC showed a minimum value when the number of factors was five. RMSEA was less than 0.05 when the number of factors was five, indicating an optimal value. Therefore, it was decided to conduct a factor analysis assuming a five-factor structure.

An exploratory factor analysis was performed using the maximum likelihood method, assuming a five-factor structure, and an oblimin solution was obtained. The goodness-of-fit indices were RMSEA = 0.03 (90% CI = 0.00–0.06) and TLI = 0.99, indicating that the data and the five-factor structure were appropriate for estimation.

Table 3 shows the factor loading matrix with a five-factor structure. The factor loadings for each variable are shown in the first factor (hereinafter referred to as "F1". The same applies to the second and subsequent factors.) F2 shows high loadings for 100 m, long jump, 400 m, and 110 m Hurdle; F3 shows high loadings for discus throw; F4 shows high loadings for shot put and javelin throw; and F5 shows high loadings for high jump and pole vault. The correlations among factors shown in Table 4 indicate negative correlations between F1 and F3, F4, and F5 and positive correlations between F3 and F4 and between F4 and F5. Based on these results, and regarding Linden [9] and Kondo [10], F1 was defined as running speed and body projection, F2 as running endurance, F3 as rotational throwing power, F4 as translational throwing power, and F5 as vertical leaping power.

## Latent rank theory

The upper limit of the number of ranks was set to 10, and the calculated information criterion was referred to (Table 5). As a result, only BIC continued to increase after showing a minimum; therefore, the LRT analysis was conducted using BIC as the reference and the number of ranks as seven.

Fig 3 shows the distribution of total points by latent rank. A one-way analysis of variance of the mean number of total points per rank showed a significant main effect (F(6, 800) =

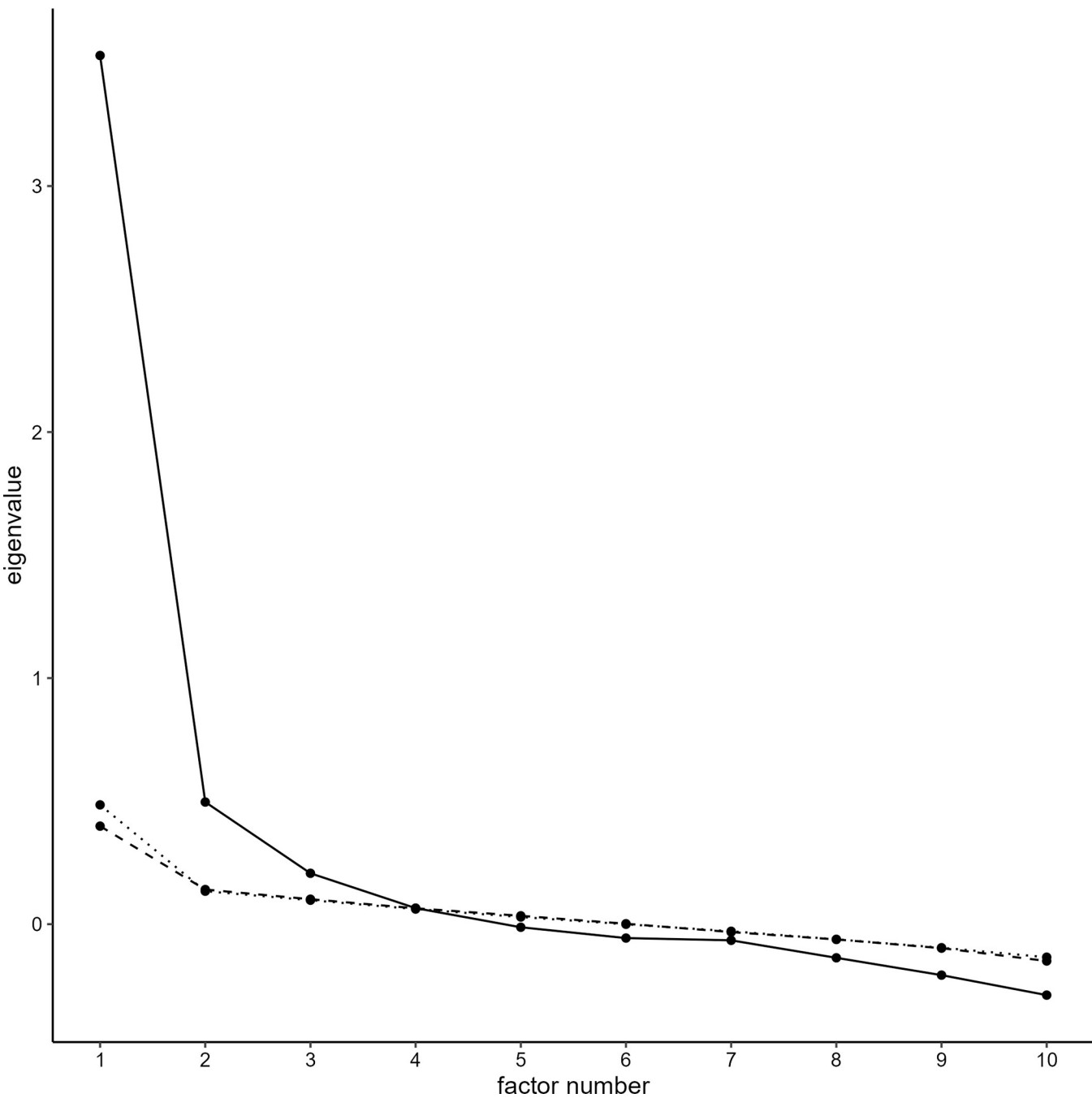

**Fig 2. Scree plots by parallel analysis.** Actual data: solid, Simulated data: dashed, Resampled data: dotted.

2128.00, p < .001, $\eta^2$ = 0.94). There was a significant difference at the 1% level between all two consecutive ranks. Then, Hedges' $g$ was calculated as the effect size. The effect size for the difference between ranks 1 and 2 was $g$ = -2.15 (95%CI = -2.53, -1.76), between ranks 2 and 3 was $g$ = -2.37 (95%CI = -2.73, -2.02), between ranks 3 and 4 was $g$ = -2.62 (95%CI -2.97, -2.28), between ranks 4 and 5 was $g$ = -2.74 (95%CI = -3.09, -2.40), between ranks 5 and 6 was $g$ = -2.99 (95%CI = -3.33, -2.65), and between ranks 6 and 7 was $g$ = -2.98 (95%CI = -3.34, -2.62). From these results, the classified ranks satisfy the ordinality for total points.

**Table 2. Model comparing AIC, BIC, and RMSEA for factor analysis of the decathlon data.**

| Factor number | BIC | AIC | RMSEA |
|---|---|---|---|
| 1 | 226.40 | 515.00 | 0.12 |
| 2 | -3.40 | 234.00 | 0.08 |
| 3 | <u>-45.60</u> | 147.00 | 0.06 |
| 4 | -34.40 | 118.00 | 0.06 |
| 5 | -24.30 | <u>94.20</u> | <u>0.03</u> |

AIC, Akaike information criterion; BIC, Bayesian information criterion; RMSEA, root mean square error of approximation.

Next, the item reference profile (IRP) for each of the 10 items and the IRP index, an index of difficulty and discrimination based on the IRP, were calculated (Table 6). Difficulty refers to the rank of the IRP closest to the theoretical median; therefore, it represents the difficulty of obtaining a high score in each category. In this study, the events with difficulty level 4 were 100 m, long jump, shot put, 400 m, 1500 m, 110 m hurdle, and javelin throw, and the events with difficulty level 5 were high jump, discus throw, and pole vault.

Discriminative power was calculated as the difference of IRP between two consecutive ranks. High-jump, 110 m hurdle, and pole vault obtained the highest discriminative power. On the other hand, the lowest discriminative power was observed for 100 m, 400 m, shot put, and discus throw.

## Discussion

### Competition performance structure

The exploratory factor analysis using the maximum likelihood method was used to obtain an Oblimin rotation. A five-factor structure was derived for the factors involving short-distance running, specifically in events such as 100m, 400m, long jump, and 110m hurdles. These

**Table 3. Factor loadings and factor correlations matrices using the ML method and oblimin solution.**

| Variables | F1 | F2 | F3 | F4 | F5 | h2 | com |
|---|---|---|---|---|---|---|---|
| 100 m | <u>0.76</u> | -0.05 | -0.12 | 0.07 | -0.09 | 0.66 | 1.1 |
| 400 m | <u>0.61</u> | 0.14 | 0.16 | -0.16 | 0.14 | 0.42 | 1.5 |
| 110 m Hurdle | <u>0.36</u> | 0.02 | -0.06 | 0.06 | -0.14 | 0.20 | 1.4 |
| Long Jump | <u>-0.53</u> | -0.01 | 0.03 | 0.13 | 0.12 | 0.49 | 1.2 |
| 1500 m | -0.01 | <u>1.00</u> | -0.02 | 0.01 | -0.01 | 1.00 | 1.0 |
| Discus Throw | -0.02 | -0.03 | <u>0.94</u> | 0.04 | 0.01 | 0.97 | 1.0 |
| Javelin Throw | -0.01 | 0.03 | 0.02 | <u>0.69</u> | 0.17 | 0.64 | 1.1 |
| Shot Put | -0.10 | -0.01 | 0.29 | <u>0.59</u> | -0.05 | 0.71 | 1.5 |
| Pole Vault | -0.04 | -0.03 | 0.03 | 0.06 | <u>0.71</u> | 0.62 | 1.0 |
| High Jump | -0.16 | -0.09 | -0.08 | 0.16 | <u>0.38</u> | 0.31 | 2.0 |
| SS loadings | 1.65 | 1.06 | 1.17 | 1.21 | 0.93 | | |
| Proportion var | 0.16 | 0.11 | 0.12 | 0.12 | 0.09 | | |
| Cumulative var | 0.16 | 0.27 | 0.39 | 0.51 | 0.60 | | |
| Proportion Explained | 0.27 | 0.18 | 0.19 | 0.20 | 0.15 | | |
| Cumulative Proportion | 0.27 | 0.45 | 0.64 | 0.85 | 1.00 | | |

ML,Maximum Likelihood.

**Table 4. Correlation matrix between each factor.**

| Factor | F1 | F2 | F3 | F4 | F5 |
|---|---|---|---|---|---|
| F1: running speed and body projection | 1 | | | | |
| F2: running endurance | 0.28 | 1 | | | |
| F3: rotational throwing power | -0.41 | 0.14 | 1 | | |
| F4: translational throwing power | 0.57 | -0.13 | 0.65 | 1 | |
| F5: vertical leaping power | -0.46 | -0.11 | 0.38 | 0.48 | 1 |

factors exhibited substantial factor loadings, aligning with the findings reported by Fan [3] and Park & Zatsiorsky [4]. Following a review of the reports using cluster analysis instead of factor analysis, we can observe that Cox & Dunn [2], which covered the World Championships from 1991 to 1999, Brodáni et al. [1], which covered only the results with more than 8000 points from 1986 to 2019, and Woolf et al. [7], which only included results above 8000 points from 1986 to 2019, also reported clusters with precisely the same structure. Although long jump is the only jumping event included in this factor, many studies have reported a significant relationship between long jump performance and sprinting ability [21–23]. In addition, 1500m constituted an independent factor in most reports [1–8]. Therefore, it is considered that two factors, "speed" and "endurance," are common in "running", regardless of the population. The term "running" should be considered separately from the physiological, morphological, and statistical aspects.

On the other hand, most of the previous studies on "throwing" have reported one factor [2, 3, 6, 7, 10, 24] in most cases, and in the case of two factors [9], only shot put throw fell under the other factors or clusters. Nonetheless, in this study, in contrast to the aforementioned reports, a unique factor emerged, comprising solely the discus throw, while another factor included either the shot put or javelin throw. This is considered a unique structure of the Japanese university student decathletes in this study. The reason that discus throw constituted a factor different from shot put and javelin throw may be due to how the horizontal speed of the projectile is acquired. The Athletics Instruction Manual [25] states that throwing events can be broadly classified into those in which the translational motion of the body obtains the horizontal speed of the projectile (shot put and javelin throw) and those in which the horizontal speed is obtained by the forward motion accompanied by the rotational motion of the body (discus

**Table 5. Model comparing AIC, BIC, and SBIC for latent rank analysis of the decathlon data.**

| Rank Number | AIC | BIC | SBIC |
|---|---|---|---|
| 1 | 113310.10 | 113413.36 | 113343.49 |
| 2 | 109506.64 | 109717.84 | 109574.94 |
| 3 | 107738.77 | 108057.91 | 107841.97 |
| 4 | 106918.09 | 107345.18 | 107056.20 |
| 5 | 106560.30 | 107095.34 | 106733.32 |
| 6 | 106392.01 | 107034.99 | 106599.94 |
| 7 | 106270.23 | 107021.16 | 106513.07 |
| 8 | 106217.61 | 107076.49 | 106495.36 |
| 9 | 106227.95 | 107194.78 | 106540.61 |
| 10 | 106159.61 | 107234.38 | 106507.17 |

AIC, Akaike information criterion; BIC, Bayesian information criterion; SBIC, Singuler Bayesian information criterion.

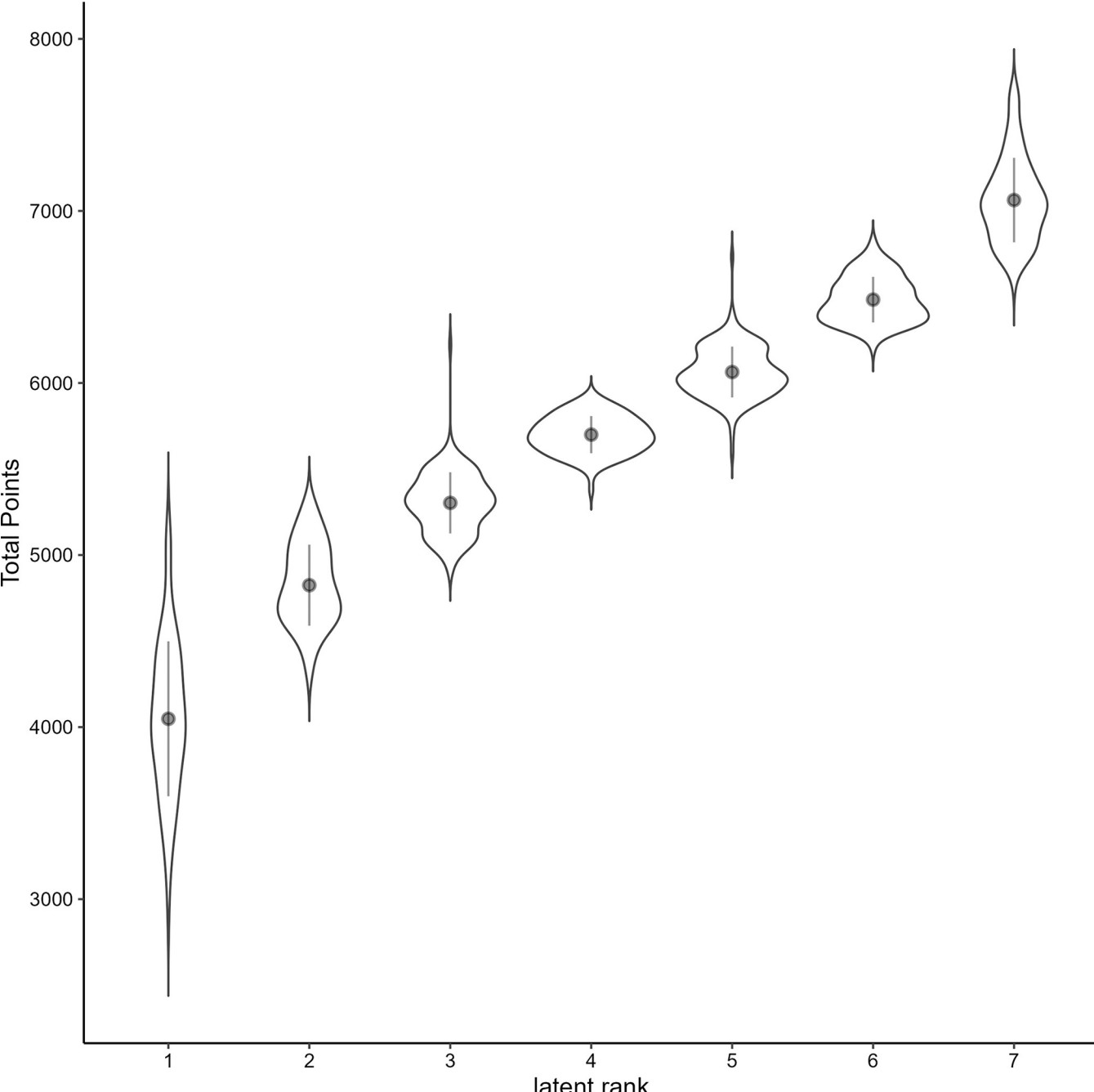

**Fig 3. Violin plot showing the distribution of overall scores for each latent rank.**

throw and hammer throw). Currently, there are two types of throws involved in the shot put. One is the "glide throw" [26], in which the body is moved from behind to in front of the circle in the following order: from a standing posture with the back to the throwing direction, the body is crouched down once, the left leg is swung backward, the right leg is kicked, and the right leg is pulled back. The other is the "rotational throw," a forward movement across the circle, like discus throw, followed by a conversion to a rotational movement [27]. While the rotational throw is the mainstream in foreign athletes, especially in the U.S., the glide throw is

**Table 6. Item reference profile (IRP) of decathlon discipline.**

| Variables | Latent Rank | | | | | | | IRP index | |
|---|---|---|---|---|---|---|---|---|---|
| | 1 | 2 | 3 | 4 | 5 | 6 | 7 | Dis. | Dif. |
| 100 m | 656.8 | 683.4 | 723.2 | 741.5 | 770.5 | 799.9 | 835.9 | 39.8 | 4 |
| Long Jump | 510.5 | 568.1 | 630.7 | 655.5 | 695.9 | 735.2 | 788.9 | 62.6 | 4 |
| Shot Put | 377.3 | 394.7 | 414.9 | 449.1 | 477.5 | 518.3 | 559.4 | 41.1 | 4 |
| High Jump | 313.6 | 480.3 | 536.0 | 560.9 | 604.1 | 638.7 | 681.6 | 166.7 | 5 |
| 400 m | 551.3 | 604.2 | 655.8 | 693.4 | 730.8 | 756.1 | 807.6 | 52.9 | 4 |
| 110 m Hurdle | 417.6 | 523.3 | 626.4 | 684.8 | 729.5 | 773.2 | 825.3 | 105.8 | 4 |
| Discus Throw | 312.7 | 334.7 | 357.0 | 385.1 | 427.7 | 476.8 | 528.6 | 51.8 | 5 |
| Pole Vault | 177.9 | 271.7 | 356.4 | 442.6 | 483.5 | 573.0 | 675.6 | 102.6 | 5 |
| Javelin Throw | 364.0 | 429.4 | 446.2 | 500.1 | 535.4 | 586.4 | 625.4 | 65.4 | 4 |
| 1500 m | 471.2 | 541.8 | 560.2 | 586.6 | 614.4 | 626.3 | 656.4 | 70.6 | 4 |

mostly used in the decathlon in Japan. Based on the above, the throwing events can be divided into two factors: rotational throwing power and translational throwing power, as a result of which the competition performance structure of Japanese university student decathletes can be interpreted.

Although "jumping" has been evaluated differently in previous studies, most of the estimates based on factor analysis tend to show that each event constitutes a different factor, and high jump tends to fall under the same factor as "running" [5, 8, 9]. However, in this study, high jump and pole vault showed high loadings on the same factor, and long jump showed high loadings on "running speed". This may be due to the difference in the direction of motion of the body's center of gravity required after the stepping-off motion in the jumping events. Ae [28] reported differences among jumping events in athletics events, focusing on the mechanism of vertical velocity acquisition of the body's center of gravity. In addition, the Japan Association of Athletics Federations [25] described the components of records in various jumping events from a biomechanical point of view. According to these, the jumping events are consistent until the athletes obtain a high sprinting velocity during the running phase and then step off at a high center of gravity. However, obtaining horizontal velocity in long jump and vertical velocity in high jump at the body's center of gravity is vital after leaving the ground. In addition, regarding pole vault, kinetic energy obtained during the run-up is temporarily stored in the pole as elastic energy, which is converted to potential energy during the pole extension. This suggests a high degree of similarity between the two disciplines. Based on the above, factor 5 can be interpreted as the vertical leaping power, in which high jump and pole vault showed high loading.

## Application of LRT

This study determined the number of latent ranks regarding three information criteria. As a result, the latent ranks were divided into seven ranks. The difference in total points among the ranks was more than 1.00 based on the effect size, $g$, which is considered a significant difference from an inferential statistical point of view. By increasing the number of ranks, we can learn more about the participant's profile. However, if the number of ranks is increased too much, the effect of the change in rank on the items becomes small and practically meaningless. In addition, the method used in this study allows the number of groups classified to be arbitrarily determined. Therefore, selecting a model that is easier to interpret and more satisfactory, based on indices, such as the information criterion, is necessary while considering the order structure of the competition level that the athletes express.

The IRP index, an accessory in LRT that helps to interpret the characteristics of each variable and latent rank, was calculated from the IRP. long jump, shot put, 100, 400, and 1500 m, 110 m hurdle, and javelin throw had the lowest difficulty index among the 10 events. Among them, 100 m, 400 m, and shot put also showed low values in discrimination (disc. = 39.77–52.90). Pavlović et al. [5] reported that speed and power are essential for transcending comprehensive motor skills in the decathlon. Therefore, shot put, 100 m, and 400 m are considered the base events in the athletic performance of Japanese university student decathletes and require high performance regardless of the level of competition. On the other hand, 110 m hurdle showed high discrimination. In the 110 m hurdle, the stride length during the race is approximately defined because the distance between the hurdles is constant [29]. Therefore, the pitch height in the sprint and the hurdling motion across the hurdles have a significant impact on the record. These results suggest that events with low relative difficulty and in which sprinting is heavily weighted, but specific technical factors play a significant role in the outcome, are likely to discriminate between the levels of competition.

The events with the highest difficulty indices–high jump, discus throw, and pole vault–showed low discriminative power among them. In particular, discus throw had little in common with the other events in the factor analysis, and it was difficult to believe that they acted as discriminative factors for the level of competition. On the other hand, high jump and pole vault showed high discriminative power. Matsubayashi et al. [30] reported on the characteristics of jumping performance among Japan's top-level decathletes, including the differences between the top-level Japanese athletes who specialize in jumping events and the top-level athletes in the world. It was noted that pole vault, not only among the Japanese top-level decathletes but also the world's top-level decathletes, had little relationship with 100 m, which is expected to reflect their sprinting ability to a large extent. Also, high jump differs from other jumping events in that it requires a curved running start. Moreover, the jumping ability required for high jump involves body rotation, swinging movements of both arms and legs, and flexion and extension of the stepping leg and trunk [28].

In conclusion, high jump and pole vault, as well as vertical leaping power, are considered to be the most complex and technical performance structures in the decathlon. Bilic et al. [31] speculated that the total points of the top-ranked decathletes in the world may be determined more by technical efficiency than by the level of basic motor skills, and the achievement level of performance in these two events can be inferred as influential factors on the level of competition in the decathlon.

In this research, the oblique solution, specifically the oblimin solution, was used for interpretability. As a result, an easily interpretable solution was obtained, however the overall factor contribution was only about 60%. Therefore, it is necessary to fully consider other factors when applying this model to actual athletes. The findings of this study also unveiled the key events that are shared across all levels of competition, as well as those events that differentiate between different levels of competition. Constructing a predictive model of athletic performance using these events is expected to lead to the discovery and development of athletes. Nonetheless, our ability to deduce the variations and impact of these results remains limited when concentrating solely on the core events and those that distinguish between levels in comparison to other events. By simulating these events, we can ascertain whether they genuinely qualify as essential events or not, and also investigate if they conform to "the principle of allocation" as described by Van Damme et al. [12].

## Acknowledgments

The results of the competitions covered in this study are publicly available on the web page of The Inter-University Athletics Union of Japan. We would like to thank Editage (www.editage. jp) for English language editing.

## Author Contributions

**Conceptualization:** Yasushi Ikuta, Noriyuki Kida.

**Data curation:** Yuki Ashino.

**Formal analysis:** Yuki Ashino.

**Investigation:** Yuki Ashino.

**Methodology:** Yuki Ashino.

**Project administration:** Yuki Ashino.

**Resources:** Yuki Ashino.

**Software:** Yuki Ashino.

**Supervision:** Yasushi Ikuta, Noriyuki Kida.

**Validation:** Yuki Ashino.

**Visualization:** Yuki Ashino.

**Writing – original draft:** Yuki Ashino.

**Writing – review & editing:** Yuki Ashino, Yasushi Ikuta, Noriyuki Kida.

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
