## [Decision Letter · Decision Letter 0]

15 Sep 2023

PONE-D-23-24889Estimation of Factors Contributing to Level Differences in Japanese University Decathlon AthletesPLOS ONE

Dear Dr. Ashino,

Thank you for submitting your manuscript to PLOS ONE. After careful consideration, we feel that it has merit but does not fully meet PLOS ONE’s publication criteria as it currently stands. Therefore, we invite you to submit a revised version of the manuscript that addresses the points raised during the review process.

 Both reviewers found your article of interest but each recommended minor revisions. Please make sure you address each of their concerns in the manuscript (e.g. put effect size calculations in the paper). In addition, please make your data available (a link to where it can be found or upload it).

We look forward to receiving your revised manuscript.

Kind regards,

Jeremy P Loenneke

Academic Editor

PLOS ONE

Journal Requirements:

Reviewers' comments:

Reviewer's Responses to Questions

**Comments to the Author**

1. Is the manuscript technically sound, and do the data support the conclusions?

Reviewer #1: Yes

Reviewer #2: Yes

2. Has the statistical analysis been performed appropriately and rigorously? 

Reviewer #1: Yes

Reviewer #2: Yes

3. Have the authors made all data underlying the findings in their manuscript fully available?

Reviewer #1: No

Reviewer #2: No

4. Is the manuscript presented in an intelligible fashion and written in standard English?

Reviewer #1: Yes

Reviewer #2: Yes

5. Review Comments to the Author

Reviewer #1: The manuscript titled "Estimation of Factors Contributing to Level Differences in Japanese University Decathlon Athletes" is well-written with good English language usage and fluent delivery. The methodology and analyses are accurate and adequately described.

Specific Comments:

1. Page 3, Line 36-38:

- The statement in this sentence appears way too strong, and it lacks a reference. In this section, the authors argue against classifying disciplines as running, jumping, and throwing. While I understand their point, I believe it's essential to distinguish between the "types of movements" (such as running, jumping, and throwing) and the physical factors influencing performance in these movements. These factors may not necessarily align with the same classification, as the authors suggested.

2. Page 3-4, Lines 40-45:

- Linden's study included Olympic-level subjects, whereas Wimmer and Kondo examined world's top-level decathletes. Are the authors considering Olympic-level athletes as world top-level in their context? Clarification on this point is needed.

3. Page 4, Lines 27-50:

- The transition in this sentence seems abrupt. The authors were discussing the study by Bilic, but then they shift to the 2023 Athletics of Japan. Additionally, it would be more informative if the focus were on the participants' performance level rather than their country of origin. For instance, in line 52, the authors mention "differences between the world's top decathletes and the top Chinese and Japanese athletes." Is this difference more dependent on the athletes' performance levels than their nationalities?

4. Methods:

- Do the authors have sociodemographic characteristics of the participants, such as age, weight, and body mass? Providing these details would enhance the description of the sample. It could be important for the reader to explicitly mention that the participants were exclusively male since decathlon is a male-only discipline.

5. Page 7, Lines 79-80:

- These lines should be integrated into the figure caption for better contextual placement.

6. Page 9, Line 112:

- Please clarify what "HAD" stands for.

7. Discussion:

- While the discussion is well-structured, I recommend including more comparisons with previous performance models/structures published in the literature. For reference, I suggest the authors review the brief communication in Nature by Van Damme and colleagues, titled "Performance constraints in decathletes." It could provide valuable insights.

8. Limitations:

- A dedicated section on limitations should be added at the end of the paper. This section should highlight the limitations of the model and possibly speculate on future directions.

In summary, this manuscript presents valuable insights into the factors influencing performance in decathlon athletes, with a focus on Japanese university decathletes.

Addressing the points mentioned in these comments will likely enhance the clarity and comprehensiveness of the paper.

Reviewer #2: Overall, from a statistical point of view, I believe the manuscript accurately describes a well thought out analysis process. The authors demonstrate a fairly simple demonstration of the use of factor analysis and latent variables for analyzing decathlete performances.

My specific comments are the following:

1. Please provide the exact effect size calculations.

2. If possible, the FigShare repository should also have the R scripts used to analyze the data.

3. I think the discussion could use some comments on how these results would compare to a similar analysis of elite (i.e., Olympic level) decathletes.

6. PLOS authors have the option to publish the peer review history of their article (what does this mean?). If published, this will include your full peer review and any attached files.

Reviewer #1: No

Reviewer #2: No

---

## [Author Response · Author response to Decision Letter 0]

27 Oct 2023

All responses to the reviewers are listed in the "Responses to Reviewers" section of the rebuttal letter.

---

## [Decision Letter · Decision Letter 1]

15 Nov 2023

Estimation of Factors Contributing to Level Differences in Japanese University Decathlon Athletes

PONE-D-23-24889R1

Dear Dr. Ashino,

We’re pleased to inform you that your manuscript has been judged scientifically suitable for publication and will be formally accepted for publication once it meets all outstanding technical requirements.

Kind regards,

Jeremy P Loenneke

Academic Editor

PLOS ONE

Additional Editor Comments (optional):

Reviewers' comments:

Reviewer's Responses to Questions

**Comments to the Author**

1. If the authors have adequately addressed your comments raised in a previous round of review and you feel that this manuscript is now acceptable for publication, you may indicate that here to bypass the “Comments to the Author” section, enter your conflict of interest statement in the “Confidential to Editor” section, and submit your "Accept" recommendation.

Reviewer #1: All comments have been addressed

Reviewer #2: All comments have been addressed

2. Is the manuscript technically sound, and do the data support the conclusions?

Reviewer #1: Yes

Reviewer #2: Yes

3. Has the statistical analysis been performed appropriately and rigorously? 

Reviewer #1: Yes

Reviewer #2: Yes

4. Have the authors made all data underlying the findings in their manuscript fully available?

Reviewer #1: Yes

Reviewer #2: Yes

5. Is the manuscript presented in an intelligible fashion and written in standard English?

Reviewer #1: (No Response)

Reviewer #2: Yes

6. Review Comments to the Author

Reviewer #1: I am satisfied with the Authors' responses to all my previous concerns. Just one note: I have noticed you have cited the work by Van Damme and colleagues as 'Van et al.', while I believe 'Van Damme' is the surname, so 'Van Damme et al.' should be the correct form.

Reviewer #2: Thank you for responding to my comments and the comments of the other reviewer. All concerns have been addressed.

7. PLOS authors have the option to publish the peer review history of their article (what does this mean?). If published, this will include your full peer review and any attached files.

Reviewer #1: No

Reviewer #2: No

---

## [Editor Report · Acceptance letter]

21 Nov 2023

PONE-D-23-24889R1 

Estimation of Factors Contributing to Level Differences in Japanese University Decathlon Athletes 

Dear Dr. Ashino:

I'm pleased to inform you that your manuscript has been deemed suitable for publication in PLOS ONE. Congratulations! Your manuscript is now with our production department. 

Kind regards, 

on behalf of

Dr. Jeremy P Loenneke 

Academic Editor

PLOS ONE